# Current Insights in Genetics of Sarcoidosis: Functional and Clinical Impacts

**DOI:** 10.3390/jcm9082633

**Published:** 2020-08-13

**Authors:** Alain Calender, Thomas Weichhart, Dominique Valeyre, Yves Pacheco

**Affiliations:** 1Department of Molecular and Medical genetics, Hospices Civils de Lyon, University Hospital, 69500 Bron, France; yves.pacheco@univ-lyon1.fr; 2CNRS UMR 5305, Tissue Biology and Therapeutic Engineering Laboratory, University Claude Bernard Lyon 1, 69007 Lyon, France; 3Center for Pathobiochemistry and Genetics, Institute of Medical Genetics, Medical University of Vienna, 1090 Vienna, Austria; thomas.weichhart@meduniwien.ac.at; 4INSERM UMR 1272, Department of Pulmonology, Avicenne Hospital, University Sorbonne Paris Nord, Saint Joseph Hospital, AP-HP, 75014 Paris, France; dominique.valeyre@aphp.fr

**Keywords:** sarcoidosis, genetics, T cell activation, mTOR, autophagy, innate immunity

## Abstract

Sarcoidosis is a complex disease that belongs to the vast group of autoinflammatory disorders, but the etiological mechanisms of which are not known. At the crosstalk of environmental, infectious, and genetic factors, sarcoidosis is a multifactorial disease that requires a multidisciplinary approach for which genetic research, in particular, next generation sequencing (NGS) tools, has made it possible to identify new pathways and propose mechanistic hypotheses. Codified treatments for the disease cannot always respond to the most progressive forms and the identification of new genetic and metabolic tracks is a challenge for the future management of the most severe patients. Here, we review the current knowledge regarding the genes identified by both genome wide association studies (GWAS) and whole exome sequencing (WES), as well the connection of these pathways with the current research on sarcoidosis immune-related disorders.

## 1. Introduction

Sarcoidosis belongs to the group of multifactorial diseases and implicates various etiological triggers, including environmental (mineral micro- and nanoparticles), infectious, and genetic factors [1,2]. This inflammatory disease is characterized by the formation and accumulation of non-necrotizing epithelioid cell granulomas in the lungs, although the skin, eyes, bones, liver, spleen, heart, upper respiratory tract, and nervous system can also be affected [3]. Extensive research over the last 25 years allowed to better understand the clinical aspects of sarcoidosis and contributed to improvement of the diagnosis, clinical follow-up, and treatment of the disease. Nevertheless, despite many studies on the biological background of inflammatory process and advances in the understanding of the putative mechanisms of granuloma formation, the etiology of sarcoidosis remains unknown. The clinical heterogeneity of sarcoidosis is significant and the phenotypes vary from asymptomatic forms, to severe and progressive disease in almost 20% of cases. The epidemiology of sarcoidosis suggests that genetic background in different ethnicities may favor the occurrence of the disease. Higher incidence rates, greater than 60 cases per 100,000, have been reported in Northern European countries, such as Sweden and Iceland, and this value is in contrast with lower incidence observed in the South European regions [4,5]. Interestingly, high incidence rates up to 30 per 100,000 have been reported in Afro-Americans in the United States, while this rate was observed to be as low as 10 per 100,000 in African individuals living in Europe [6]. Although these different studies may be biased by the diagnostic criteria for sarcoidosis, which optimally requires the detection of the granuloma in a lung biopsy, the ethnic and geographic variations in the incidence values strongly suggest the entanglement of genetic and environmental factors inducing the disease. The existence of familial forms of sarcoidosis suggests that genetic predisposition plays an important contributory role in sarcoidosis pathogenesis in humans [7]. The sarcoidosis genetic linkage study consortium (SAGA) established to recruit African-American affected sib pair families identified 229 families, with only 15 percent having three or more affected sibs, and allowed the identification of eight chromosomal regions with suggestive evidence for linkage to sarcoidosis susceptibility [8,9]. The largest multicenter study, ACCESS (A Case-Control Etiologic Study of Sarcoidosis) confirmed the possibility of familial clustering of sarcoidosis, supporting an inherited susceptibility to sarcoidosis. Siblings of patients with sarcoidosis were shown to have around a fivefold increased risk of developing sarcoidosis [10]. A recent case–control–family retrospective study nested in Swedish population-based registers (Swedish Research Council) showed that having a relative with sarcoidosis is a risk factor and estimated the mean heritability of sarcoidosis at 39% [11]. This precise study is in agreement with several previous works showing that sarcoidosis can occur in a hereditary context with a mode of transmission suggesting the effect of dominant mutations. Despite many years of genetic association studies on large cohorts of sarcoidosis patients, no major genes have been implicated to this predisposition with a significant pathogenic link with disease occurrence. However, the arrival of new high-throughput sequencing technologies (NGS) very recently contributes to the identification of major genetic factors, which could explain the family predisposition and help the understanding of the mechanistic background of granuloma formation and sarcoidosis occurrence.

## 2. Genome Wide Association Studies (GWAS) in Sarcoidosis

### 2.1. GWAS: from Success to Disappointments

Genetic loci linked to specific phenotypic traits have largely been determined by genome-wide association studies (GWASs) in many multifactorial common and rare diseases [12]. The principle of GWAS is to compare and associate millions of relatively common genetic variants, usually single-nucleotide polymorphisms (SNPs), between a baseline (control) population and one with a well-defined phenotype, and in this case, a disease such as sarcoidosis for which the classical techniques of candidate gene and/or approach by reverse genetics (linkage analysis in predisposed families) are difficult to apply. The GWAS technique can thus be compared to jumps from SNP to SNP over distances of variable size, hence the need to reinforce the density and the number of polymorphic markers used. Each marker-variant contributes to establish a haplotype in a specific region, a so-called candidate locus, containing many nearby variants that are in high linkage disequilibrium (LD), because that they are most likely to be inherited together. GWAS is suitable for common and complex diseases in terms of mechanism, such as psychiatric conditions, diabetes, and many inflammatory and/or autoimmune and inflammatory diseases, and particularly sarcoidosis [12,13]. However, it is important to remember the fundamental criteria for giving the results of GWAS an unbiased statistical value: (a) the density of markers, and thus their number, must be large, greater than 500,000 distributed over the genome; (b) the clinical definition of the disease studied must be precise in order to avoid the integration of phenocopies (a phenotype mimicking the disease) and, in sarcoidosis, the ideal situation would be for all the diagnoses to be confirmed by histopathological examination; (c) the number of patients and control subjects included in the study must be as large as possible, which represents a major difficulty in the absence of a national or international clinical network; and (d) the statistical interpretation of the results in a candidate region of the genome must take into account the frequency of alleles of the SNP markers of this locus in the normal population. Despite the method’s great initial promise, GWASs have not provided immediate knowledge on the underlying biological mechanisms of many complex diseases. The reasons are firstly that GWAS cannot distinguish a marker-variant signal from that of the other variants that are in higher LD and not included in the SNP panel. Most of the SNPs identified by high LD are located outside (up to 500–1000 Kb) of coding sequences of the genes and do not allow a functional interpretation [12]. Second, in most cases, GWAS highlights a genomic region and not a candidate gene nor a pathogenic mutation. Further fine-mapping analysis will be needed to identify putative candidate genes and, for most SNPs, they may affect regulatory regions close to or far away from genes, making functional interpretation very difficult. A similar conclusion may be drawn for SNPs located inside and mainly in the middle part of introns and putatively disturbing the regulatory intronic splicing enhancers (ISEs) and silencers (ISSs) [14]. Nevertheless, GWAS provide interesting data in several diseases and in particular for sarcoidosis, but these results must be assessed by complementary fine-mapping approaches that have been developed in what is called the post-GWAS era with the aim of identifying the important disease-causing or risk-increasing genes and interpreting their biological impact leading to the disease occurrence [15,16].

### 2.2. Contribution of GWAS to Understanding Immunogenetics of Sarcoidosis

As introduced previously, variations in susceptibility to sarcoidosis in different ethnic groups, familial clustering, and increased concordance in monozygotic twins are the three main observations suggesting that a genetic background may act synergistically to environmental factors for triggering the disease [16]. Genome-wide association studies provided a significant number of candidate loci, some of the genes that have been studied functionally to give functional meaning to genetic observation. The most important genetic risk factors have been identified at 6p21, a locus containing major antigen presentation and T-cell regulation related genes, such as human leukocyte antigen (HLA) and *BTNL-2* (Butyrophilin-like 2 protein), a cofactor of CD80/CD86 acting as a negative regulator of T-cell activation during the presentation of antigens by APCs (antigen presenting cells) [17,18,19]. This interesting observation was highlighted by the fact the main *BTNL-2* SNP related to sarcoidosis, rs2076530, induces the activation of a cryptic splice site located 4 bp upstream of the exon five-intron wild-type donor splice-site site and a premature a premature stop in the spliced mRNA [20]. The resulting protein lacks the C-terminal IgC domain and transmembrane helix, thereby disrupting the membrane localization of the protein. It could thus be considered that this mutation releases the extracellular part of *BTNL-2* protein and disturbs the regulation of T-cell activation. This is probably one of the significant success stories of GWAS in sarcoidosis allowed by the analysis of 947 independent cases of familial and sporadic sarcoidosis. Nevertheless, the *BTNL-2* rs2076530 (G16071A) variant is commonly observed in the baseline population, with a mean minor allele frequency (MAF) of 0.4194 (41.94%), varying from 0.3304 (33.04%) to 0.4596 (45.96%) in African and European populations, respectively [21,22]. This means that a common polymorphism may be related to a rare disease. The odds ratio (OR) associated with rs2076530 was estimated at 1.8 to 2.0 in several studies, but we demonstrated in a French cohort that this *BTNL-2* variant could not explain the autosomal dominant predisposition to sarcoidosis in affected families [23]. However, *BTNL-2* remains an interesting candidate gene that may explain a basal predisposition to the disease, but requiring additional genetic factors until reaching a threshold of predisposition, a prerequisite for inducing the abnormal accumulation of granuloma.

An extensive description of the *BTNL-2* history is a critical step for understanding that sarcoidosis, as many multifactorial and common diseases, has a complex genetic architecture and, as a result, DNA analysis to predict disease will implicate a panel of genes and the establishment of a polygenic risk score to identify patients who could benefit from the knowledge of their susceptibility to the disease, but also those with particular and/or severe forms of the disease. In sarcoidosis, specific class HLA class II antigens have been associated with an increased risk of disease occurrence and some of them with certain sarcoidosis phenotypes [24,25]. The ACCESS project (A Case Control Etiologic Study of Sarcoidosis) was the first to show that HLA-DRB1 (*1101) and HLA-DPB1 (*0101) alleles contributed significantly to the risk of sarcoidosis, in both black and white populations [26]. Regarding the genotype–phenotype correlations, HLA-DRB1 * 14, DRB1 * 15, and DQB1*0601 haplotypes are associated with chronic forms of sarcoidosis [27,28]. An acute, but resolving expression of the disease, so-called Lofgren syndrome, appears more frequently with the HLA-DRB1*03 and DQB1*0201 alleles [29,30]. As discussed by several authors, the diagnostic value of HLA information is more focused on helping with phenotypic classification of patients and having a relative predictive value of the prognosis and/or specific organ localization of sarcoidosis. Disturbingly, however, the interpretation of HLA alleles may differ between ethnic groups such as, for example, in the Han Chinese population, where HLA-B * 51 has been shown to be more associated to Lofgren syndrome instead of DRB1*03 [31]. Various organs’ involvement have been shown with specific HLA alleles, such as, for example, a decreased risk of extra pulmonary manifestations in DRB1*0301 carriers and an increased risk of extra thoracic and skin lesions in DRB1*04 and DRB1*0302 carriers, respectively [32,33]. Taken together, HLA class II haplotypes are now considered as strong genetic determinants involved in sarcoidosis predisposition. The HLA-DRB1 * 04 allele has been shown to increase the risk of sarcoid uveitis, a major complication of the disease, in English and Japanese groups of patients, and more recently in a large GWAS study focusing on ocular sarcoidosis in both European and African-American cohorts [27,34]. An association between the HLA-DQB1*0601 allele and cardiac sarcoidosis was also identified in the Japanese cohort [35]. Interestingly, it is well-known that exposure to mineral and nanoparticles (NNPs) during the NYC World Trade Center disaster in September 2001 has been a predisposing factor to “sarcoidosis-like” granulomatous pulmonary disease among New York Fire Department rescue workers [36]. Even in this highly environmental inducing context, it was recently shown that more that seventeen novel HLA alleles were associated with an increased risk of sarcoidosis in exposed individuals, with OR values varying from 1.66 (HLA-DPA1/HLA-DPB1) to 5.21 (HLA-DRB1) [37].

Behind the complexity of the HLA genes allelic combinations and the relative risks involved, interesting observations link sarcoidosis to infectious agents and autoimmunity. The immune response to mycobacterial antigens has been shown to be dependent on DRB1 haplotypes, with a more significant association to DRB1*0301, the one associated with Lofgren syndrome and resolving disease [38]. The production of auto antigens in an HLA-DRB*0301 positive patient has been highlighted in a Swedish study by the observation of a particular CD4+ T cell subset, which recognizes inherent peptides such as vimentin and ATP synthase [39]. These two examples specifically suggest that the structure of HLA molecules and the prediction of the possibilities of attachment to peptides and other antigens is a determining factor in the ability of an individual to eliminate the infectious or non-infectious agents to which he is subjected in his environment.

Within the region of the major histocompatibility complex and between the regions of class I and class II genes, several genes have been described that appear to be involved in both global and specific inflammatory responses. Immune-related functions are characteristic to molecules belonging to major histocompatibility complex (MHC) class III as genes for some complement proteins, cytokines, and heat shock proteins lie in the region [40]. Among these loci, *BTNL-2* was the most extensively analyzed in sarcoidosis. Other genes, such as *NOTCH4* (neurogenic locus notch homolog 4), *TAP2* (transporter 2, ATP binding cassette subfamily B member), TNFα (tumor necrosis factor α), *LTA* (lymphotoxin α), *HSPA1L* (heat shock 70 kDa protein 1L), and several open reading frames have been putatively associated with the immune process involved in the disease [29,41]. A specific polymorphism in *NOTCH4* (rs715299) has been shown to be associated with sarcoidosis in African Americans and European patients independently from others in the MHC region in the same sample [40]. To date, this intronic variant with a very low MAF value (<0.00003) has not been related to a dysfunction of the *NOTCH4* protein, but Notch family proteins are involved in regulation of T-cell immune response and are related to other autoimmune diseases such as multiple sclerosis and lupus [41,42]. *TAP2* (transporter 2, ATP binding cassette subfamily B member) moves protein fragments from foreign invaders into the endoplasmic reticulum in close connection with major histocompatibility complex (MHC) class I proteins and *TAP2* polymorphisms have been highlighted in European cases of sarcoidosis [43,44]. Two polymorphisms of TNF-α (tumor necrosis factor alpha) are associated with susceptibility to sarcoidosis in European population [45]. Both are located in regulatory regions of the gene and, strikingly, one of them, –308 A > G, was shown to decrease the response to TNF inhibitors (adalimumab or infliximab) in patients with refractory sarcoidosis [46,47]. Paradoxical reactions have been described with anti-TNF therapy in various inflammatory diseases and sarcoidosis-like lesions are increasingly reported during anti-TNF treatment [48]. Interestingly, monocytes of sarcoidosis patients show an enhanced ability to aggregate in vitro after stimulation with TNF-α and interferon-β, an observation that places TNF in an ambiguous position between one of the factors contributing to the formation of the granuloma or a therapeutic target [49]. LTA, formerly known as TNFβ (tumor necrosis factor β), is expressed by CD4+ T helper type 1 (Th1) cells, CD8+ cells, natural killer (NK) cells, B cells, and macrophages, and binds with high affinity to TNF receptors [50]. A variant in LTA intron 1 was associated with erythema nodosum in female Caucasian sarcoidosis patients [51]. Lymphotoxin α may be associated with the pathogenesis of several inflammatory diseases and has been shown to be required for granuloma formation and resistance to Mycobacterium, Leishmania, and Plasmodium infections in mice [52,53].

### 2.3. Other Pathways Identified by GWAS in Sarcoidosis Etiology

Polymorphic variations in genes associated with MHC (major histocompatibility complex) probably generate a background of genetic predisposition for the development of sarcoidosis, but other GWAS studies performed identify a set of genes and loci that may contribute synergistically to the genesis of the disease. We may imagine a complex polygenic model in which the progressive accumulation of mutations in various genes increases the relative risk of the onset of sarcoidosis (Figure 1). Various pathways have thus been suggested to be involved in sarcoidosis.

One of the most relevant associations targets *ANXA11* (Annexin A11), a gene located on chromosome 10q22-23 and implicated in several biological pathways, including apoptosis and proliferation [54,55]. GWAS identifies a pathogenic SNP, rs1049550 (ANXA11), as a significant risk factor in a large German cohort of patients. Nevertheless, the mean minor allele frequency of rs1049550 is high in the European population (0.4164 or 41.64%), which suggests that ANXA11 is a necessary, but not sufficient factor for generation of the disease. Additional studies have been performed in African American patients by checking 209 SNPs in the ANXA11 gene, and found that new rare variants are independently associated with the risk of disease susceptibility and radiographic stage, while some others have a protective effect, and finally suggesting that the relationship between genetic variants in ANXA11 and sarcoidosis risk is more complex among people of African origin [56,57]. A similar observation was done recently in a cohort of 103 Greek sarcoidosis patients [58]. A non-coding SNP (rs61860052) of Annexin A11 has been associated with an increased risk of sarcoidosis-associated uveitis and other variants with pulmonary fibrosis in African American patients [59]. Annexins (ANX) are a large family of calcium-dependent membrane-binding proteins that play a critical functional role in the cell life cycle, exocytosis, and apoptosis. The ANXA11 rs1049550 causes a substitution of arginine with cysteine at position 230 (R230C), and has been suggested to induce a defective apoptosis by disturbing the annexin-mediated delivery of calcium ions necessary for the activation of caspases via mitogen-activated protein kinase (MAPK) [54]. This interesting functional relationship to sarcoidosis has been highlighted by the fact that annexins interact with S100 proteins, which are involved in many Ca^2+^-dependent processes and mainly apoptosis [60]. Interestingly, annexin A11 has a long hydrophobic N terminal sequence binding members of the S100 proteins family, such as S100A6 (calcyclin), and whose mutations have been linked to amyotrophic lateral sclerosis (ALS), a neurodegenerative disease characterized by defective autophagy and intracellular vesicular traffic in neuronal tissues [61,62]. S100 antigens are detected at a high level in the early stages of sarcoidosis granuloma formation [63]. Taken together, these data may suggest that the pathogenic effect of *ANXA11* R230C variant might be owing to a defective apoptosis and/or autophagy inside granuloma, thus inducing a persistent inflammatory reaction and a higher risk of onset of sarcoidosis and poorer outcome [64].

Another interesting gene identified by GWAS was *CCDC88B* (coiled-coil domain-containing protein 88B) located on chromosome 11q13, which encodes a positive regulator of T-cell maturation and inflammatory function involved in inflammatory bowel diseases [65,66]. Together with other 11q13.1 loci, the rs479777 SNP of CCDC88B was significantly associated with sarcoidosis in a German study. Loss of CCDC88B protein expression impacted T cell maturation and reduced activation and cytokine production (IFN-γ and TNF) in response to T cell receptor engagement [67].

Three SNPs (rs7517847, rs11465804, and rs11209026, respectively) are located in the *IL23R* gene, encoding the receptor of interleukin 23, which is involved in differentiation of the differentiated Th17 T cell population, and the development of autoimmune and inflammatory disorders has been identified as susceptibility variants for sarcoidosis [68]. Two of them, rs11465804 and rs11209026, were associated with sarcoid uveitis, with rs11209026 being a missense variant (R381Q) in exon 9 of the gene. Two scientific aspects are interesting in the study of interleukin 23 and its receptor: first, IL23 preferentially drives the developing Th17 cells into the highly inflammatory pathogenic subtype and has been associated with active Vogt‒Koyonagi‒Harada and Behçet’s uveitis [69,70]; and second, IL23R polymorphisms could be one of the factors involved in the complex interactions between smoking and the risk of developing sarcoidosis, a subject of controversy as studies show a reduction in the morbid risk in smokers [71]. Interestingly, IL23 is a member of the IL-12 (interleukin 12) family cytokines and heterodimeric IL-23 comprises a p19 helical-bundle subunit (IL-23p19), which is disulphide linked to a p40 subunit (IL-12p40), encoded by the IL12B gene [72]. IL-23 signals via its specific interleukin-23 receptor (IL-23R) and interleukin-12 receptor subunit β1 (IL-12Rβ1), which is also utilized by IL-12 [73]. In a large GWAS study, six polymorphisms of IL12B yielded a significant association with sarcoidosis and one of them (rs4921492) more specifically with central nervous system involvement [21]. IL-12 drives the differentiation of naive T cells into interferon-γ (IFN-γ)-producing T helper 1 (Th1) cells and, without a doubt, research into therapies targeting the IL12/IL23 signaling pathway, such as for anti-IL-12p40 ustekinumab, a promising treatment in psoriasis and Crohn disease, but unsuccessful in sarcoidosis, will be important for the management of patients sharing specific polymorphisms in these genes [74].

Various SNPs located near the nuclear factor kappa B subunit 1 (NFKB1) and MANBA (beta-D-mannoside mannohydrolase) genes on chromosome 4q24 showed a significant association with sarcoidosis, but no functional relationship has been established despite the well-known role of NF-kappa-B, a pleiotropic transcription factor in a vast array of stimuli related to inflammation, immunity, differentiation, cell growth, and apoptosis [18,21]. NFKB1 encodes a 105 KD protein, a Rel protein-specific transcription, and IL12B gene induction by lipopolysaccharide in bone marrow-derived macrophages requires Rel [75]. As suggested in this large GWAS study, a striking result is that some IL12B polymorphisms might modify the RelA binding site involved in the regulation of IL12B by NF-kappa-B.

A very common variant (rs3184504) of the SH2B3 (SH2B adaptor protein 3) on chromosome 12q24.12 has been considered as a susceptibility factor for sarcoidosis [21]. Despite a mean MAF value of 0.668 (66.8%) and an in silico benign predictive effect on the protein (amino-acid change: W262R), SH2B3 was considered as an interesting candidate because of its function as a negative regulator of TNF signaling; modulator in integrin signaling and actin cytoskeleton organization; and genetic link to myeloproliferative, autoimmune, and inflammatory disorders [76].

The cytoskeleton, which includes microtubules and actin filaments, is crucial for many cellular processes, such as proliferation and differentiation. An interesting association has been shown in a German study between the RAB23 gene on chromosome 6p21 and sarcoidosis, focusing on a non-synonymous SNP variant, rs10484410 (G207S) [77]. RAB23 polymorphisms have been further linked to an increased risk of sarcoidosis-associated uveitis [59]. Rab GTPases are vesicle-trafficking proteins that mainly localize to membrane-bound compartments and act as molecular switches that transform between a GTP-bound state (active form) and a GDP-bound state (inactive form) [78]. Rab GTPases participate in budding, uncoating, motility, and fusion of vesicles and play an important role in antibacterial defense through participation in autophagosome formation at the initial stages of autophagy [79]. Ocular involvement in sarcoidosis has been highlighted recently in a large GWAS screening identifying seven variants in the MAGI1 (membrane associated guanylate kinase with inverted structure 1) gene, encoding an adaptor protein that stabilizes epithelial and endothelial cell–cell contacts mainly through the PTEN/phosphatidylinositol-3 kinase/Akt and β-catenin pathways [34]. MAGI1 has been related to tumor progression in various cancers and to autoimmune diseases in rat models and in human psoriatic arthritis [80].

Among the studies using the GWAS technique, paradoxical results have been obtained on the role of NOD2 (nucleotide-binding oligomerization domain-containing protein 2) in sarcoidosis [81]. NOD2 protein, an intracellular pattern recognition receptor, is involved in recognizing certain bacteria and stimulating the immune system to respond appropriately. When triggered by specific substances produced by bacteria, the NOD2 protein activates NFκB (nuclear factor-kappa-B) through an intracellular signaling cascade implicating RIPK2 (receptor-interacting serine/threonine-protein kinase 2) and the TAK1 (TGF-beta activated kinase 1 or MAP3K7)–TAB1 (TAK1 binding protein 1)–TAB2 (TAK1 binding protein 2) complex [82]. The NOD2 protein also appears to play a role in autophagy to surround and destroy bacteria, viruses, and other harmful substances [83]. The NOD2 pathway is known to be involved in macrophage activation, and deletion of *TAB1*/*TAB2*, defined as TAK1 (TGFβ-activated kinase activators) binding proteins 1 (TAB1) and 2 (TAB2), results in macrophage death [84]. *NOD2* germline mutations are associated with Crohn disease and early onset sarcoidosis or Blau syndrome, a rare systemic inflammatory disease characterized by early onset granulomatous polyarthritis, uveitis, and dermatitis [85,86]. Studies targeting the NOD2 gene have identified mutations of *NOD2* in sarcoidosis, sarcoid-related uveitis, and orofacial granulomatosis, including those associated with Blau syndrome, suggesting that NOD2-related pathways could play an important role in basal predisposition to the risk of occurrence of uncommon forms of sarcoidosis [87,88,89]. This topic will be discussed in the next section.

Other genes have been suggested as susceptibility factors for sarcoidosis from single, unreplicated association studies. Among these, the XAF1 gene (X-linked inhibitor of apoptosis or XIAP associated factor 1) located on chromosome 17 shares an interesting intronic variant (rs6502976) in an African American cohort of patients [90]. Functional studies have shown that XAF1 induces autophagy through upregulation of Beclin 1, a key regulator of autophagosome formation, and cooperates with interferon regulatory factor (IRF)-1 to promote apoptosis under various stressful conditions such as inflammation, viral infection, and host defense [91,92]. Polymorphisms in *ZNF592* were found to be associated with neurosarcoidosis in African American and European American patients [93]. Homozygous mutations in ZNF592 have been related to CAMOS (cerebellar ataxia with mental retardation, optic atrophy, and skin abnormalities) syndrome, a rare autosomal recessive syndrome characterized by a congenital cerebellar ataxia associated with mental retardation, optic atrophy, and skin abnormalities [94].

Polymorphisms of CCR5 (C-C chemokine receptor type 5), predominantly expressed on T cells, macrophages, and dendritic cells, have been identified in a Czech series of patients, and further confirmed as being more specifically associated with lung disease progression [95,96]. CCR5 is the ligand for several cytokines, including CCL3, CCL4, and macrophage inflammatory proteins (MIPs), and interacts with CCL5 (RANTES or regulated on activation, normal T cell expressed and secreted). CCR5 is extensively studied because it is known as the co receptor of HIV1 (human immunodeficiency virus) and plays a critical role in inflammatory responses to infection [96,97]. Interestingly, polymorphisms of CCR5 have been shown to be associated with chronic beryllium disease (CBD), a granulomatous lung disease occurring in 2 to 5% of beryllium-exposed workers [98]. CBD may be clinically indistinguishable from pulmonary sarcoidosis at the time of initial diagnosis. A recent French report highlights the complexity of the diagnosis of CBD owing the similarities with sarcoidosis. Nevertheless, unlike sarcoidosis, CBD is driven by the expression of specific HLA-DBP1 alleles and the disease may be diagnosed in beryllium-exposed patients using immunologic testing, such as the beryllium lymphocyte proliferation test [99]. CD4+ T cells in the bronchoalveolar lavage (BAL) of CBD and sarcoidosis patients are highly Th1 polarized, but another difference between the two diseases is the absence of significant IL-17 producing Th17 in CBD [100]. Lastly, HLA-DPB1 with a glutamic acid at amino acid position 69 (Glu69) (or HLA-DPB1*02:01) allele confers an increased risk of beryllium sensitization and characterized CBD as a separate entity from sarcoidosis, which can be named a metal-induced granulomatous disorder [101].

## 3. Next Generation Sequencing Contributes to Genetic Research on Sarcoidosis

Taken together, GWAS studies were able to identify minor and/or major genetic factors of susceptibility to sarcoidosis, but were unable to identify strong genetic triggers, which may explain the occurrence of familial forms of the disease. As shown in the Swedish study including 23,880 participants with sarcoidosis, the relative risk associated with having at least one first-degree family member with sarcoidosis was estimated at 3.73, and increased to 4.69 if at least two first-degree family members were identified as having sarcoidosis [7]. Compared with the general population, there was an 80-fold increased risk of developing sarcoidosis in co-twins of affected monozygotic brothers or sisters [102]. The French national program of the GSF network (Group Sarcoidosis France) for ten years has focused on the familial forms of sarcoidosis (the SARCFAM project) and initially excludes the BTNL2 gene as a major predisposing factor to the disease [23]. To overcome the issue of missing heritability, next-generation DNA sequencing (NGS) techniques have been applied recently to various clinical situations, including familial and sporadic forms of sarcoidosis. Technical advances to assess rare variation genome wide, particularly whole exome sequencing (WES), have facilitated collaborative studies resulting in novel disease gene discoveries, mainly for complex diseases [103]. The main difficulties induced by NGS methods are (1) the large number of variations identified and difficulties of classification of variants between pathogenic and benign; (2) the need for a bio informatics analysis to select the genes and mutations that are potentially causal; and (3) the need to analyze the data not gene by gene, but rather in functional networks, to target the etiology of polygenic diseases.

### 3.1. WES Screening of Familial and Sporadic Forms of Sarcoidosis

Genetic screening of a complex disease by WES may be firstly done for the search of de novo mutations by analyzing a classical familial situation, the so-called trio, in which a child, despite healthy parents, develops the disease at an early age. Subtraction of parental genotypes from the affected child allows the identification of new mutations putatively linked to the disease. This protocol has been applied in three trios identified by the French GSF and RespiRare networks and allowed a focus on a series of genes sharing de novo and recessive mutations in affected children [104]. This study, unique in the world to date, identified 37 genes sharing coding variants occurring as either recessive mutations in at least two trios or de novo mutations in one of the three affected children. The genes were classified according to their potential roles in immunity related pathways: nine to autophagy and intracellular trafficking, six to G-proteins regulation, four to T-cell activation, four to cell cycle and immune synapse, and two to innate immunity. An in-depth focus on 10 of these 37 genes may suggest that the formation of granuloma in sarcoidosis may result from combined deficits in autophagy and intracellular trafficking (ex: Sec16A, AP5B1, and RREB1), G-proteins regulation (ex: OBSCN, CTTND2, and DNAH11), T-cell activation (ex: IDO2 and IGSF3), mitosis, and/or immune synapse (ex: SPICE1 and KNL1).

This first study was followed by a WES screening of five families sharing at least two first-degree affected relatives and comparing the genotypes of patients to intrafamilial healthy controls [105]. Analysis of exome data allowed the authors to selected rare and low frequency variants with minor allele frequency lower than 0.05, and short-listed those with mutations suspected of a deleterious effect by SIFT [106] and/or POLYPHENv2 [107], or those suspected of a direct or indirect effect on splicing (Alamut Visual©). This resulted in a list of 227 susceptibility variants in 195 genes after pooling all the rare variants, which were further analysed using STRING (https://string-db.org/) protein network, mapped to pathway gene sets curated in the Wikipathway 2016 database and submitted to an additional gene set over-representation pathway analysis using ConsensusPathDB, a meta-database that integrates different types of functional interactions from multiple data resources [108]. From this analysis, autophagy was found to be the top pathway enriched among the transcripts affected by the selected variants, followed by phenylethylamine degradation I, and target of rapamycin (TOR) signaling. Accordingly, the authors found mTORC1 and mTOR complex (*MTOR*, *RICTOR*, and *MLST8*) to be the top three protein complexes that encode the variant-affected transcripts. While this familial analysis confirms the genetic heterogeneity commonly reported in sarcoidosis, it allowed to stratify heritable pathogenic variants in mTOR and Rac1 signalling pathways, as well as in autophagy and vesicular transport processes. These data were highlighted by one of the critical mouse models of granuloma induced by the deletion of Tsc2 (tuberous sclerosis 2), a negative regulator of the metabolic checkpoint kinase mTORC1 belonging to the mTOR complex [109]. Deletion of Tsc2 was sufficient to induce hypertrophy and proliferation, resulting in excessive granuloma formation in vivo related to mTORC1 activation, leading to increased glycolysis and inhibition of NFκB signaling and apoptosis. These effects were reverted by inhibition of mTORC1 and similar patterns of mTORC1 activation were reported in a significant subset of granulomatous lesions in human sarcoidosis patients. A recent Chinese WES report in a single family identified deleterious variant in the ZC3H12A (Zinc finger CCCH-type containing 12A) gene, also named MCPIP1 (monocyte chemoattractant protein-1), encoding Regnase-1, a transcription factor regulation purine and energy metabolism through mTORC1 signaling, and negatively regulating the differentiation of Th17 cells, a crucial immune cell in the onset of sarcoidosis [110,111,112]. A German study performed WES screening of 22 sarcoidosis cases from six families without subtraction of intrafamilial control genotypes [113]. Similar selection procedures of variants were applied and the data analysed from both linkage chromosomal regions previously known in sarcoidosis and functional properties. The authors highlight gene variants in genes encoding regulators of pro-inflammatory cytokines, production of IFN-γ, anti-inflammatory cytokine IL-10, leukocyte proliferation, bacterial defence, and vesicle-mediated transport. Part of the data observed in the French study overlaps with genes from the WES analysis in German families, highlighting enrichment in pathways related to cell survival and migration, calcium metabolism, as well as cell adhesion processes putatively involved in immunity [104,105,113].

A recent WES analysis focused a Finnish series of 72 patients selected based on disease activity, half resolved and half persistent, in order to identify prognosis genetic markers [114]. This work used an original stratification based on the HLA haplotypes, those which may influence disease occurrence and progression (HLA-DRB1*03:01 and HLA-DRB1*04:01-DPB1*04:01), and a replication study in a large set of sarcoidosis patients and controls. Two genetic locations were shown to have significant association, one in 1p36.21, containing two genes (AADACL3 and C1orf158) associated with disease resolution independent of the HLA markers, and the second in 19q13.42 (containing LILRB4, KIR3DL1/KIR3DS1, and LAIR1). The 1p36 chromosomal region was highlighted in both the German and Finnish studies [113,114].

### 3.2. Are We Able to Integrate All the Results from the Genetic Studies?

All the genetic data confirm the very strong genetic heterogeneity in sarcoidosis. Our first impression is that we must probably distinguish the familial and sporadic forms and, in each of these groups, the particular clinical expression of the disease. The challenge of genetic studies is to identify biological markers for early diagnosis of the disease and, more reasonably for sarcoidosis, to define the genetic profiles that may be associated with severe and persistent disease, and more specifically with multi-organ lesions. We recently hypothesized that genetic mutations observed in familial sarcoidosis mostly occur on autophagy-related genes and two regulatory hubs, mTOR and Rac1, which can hamper the clearance of pathogens (opportunistic viruses or bacteria) or non-organic particles and alter both macrophages and T cell responses [115]. Rac1, a member of the Rho family of GTPases, is a critical regulator of mTOR complex (both mTORC1 and mTORC2 components) in response to growth-factor stimulation and binds directly to mTOR and mediates mTORC1 and mTORC2 localization at specific membranes [116]. Rac1 is one of the main targets of the immunosuppressive compound azathioprine (AZT) used in corticosteroid resistant forms of sarcoidosis [117]. mTOR plays a fundamental nutritional role during intracellular vesicle trafficking and negatively regulates autophagy [118]. mTOR inhibitors such as sirolimus are used in very rare and specific conditions and are shown to be successful after intravitreal injection in non-infectious uveitis [119]. Any alteration of genes encoding Rac1 or mTOR regulators, or intrinsic to autophagy, can hamper the clearance of pathogens (opportunistic viruses or bacteria) or non-organic particles and alter both macrophages and T cell responses (Figure 2). A recent report in a single case of familial sarcoidosis carrying the NOD2 G908R mutation, well-described in Crohn disease, was informative in that two pathogenic mutations in Rac1 regulators, KALRN (Kalirin) and EPHA2 (ephrin receptor A2), respectively, discriminate patients with sarcoidosis from intrafamilial NOD2 G908R + without the disease [87]. Additionally, a third mutation was observed in the interleukin 17A receptor (IL17RA). IL-17 has a broad pro-inflammatory potential in mammalian host defense, in inflammatory disease, and in autoimmunity. In a model of human alveolar macrophages, IL-17A can partially inhibit the release of IL-23 protein, considered as the upstream regulator of IL-17A, and this effect is mediated by Rac1 [120]. Furthermore, the GEFs (guanine exchange factors) regulators of Rac1 such as TIAM1 have an important function in the production of IL17 cytokine by binding to the promoter regions of the *IL17* gene within a complex containing RORγt (retinoic acid-related orphan receptor gamma), the master transcription factor of Th17 cells promoting Th17 cell differentiation [121]. As for Rac1 mediators, we can hypothesize that genetic variations observed in IL17 receptors and/or in IL12B, encoding the common p40 subunit of IL12 and IL23, could disturb the expression of IL17 and be a predisposing state for sarcoidosis.

Variations in the direct regulation of mTOR probably have major consequences on autophagy. In two families with a severe form of sarcoidosis, we demonstrated deleterious mutations in *DDIT4* (DNA damage inducible transcript 4 gene), also called Rtp801, encoding a factor that turns off the metabolic activity triggered by mTOR by stabilizing the TSC1–TSC2 inhibitory complex [105]. Interestingly, *DDIT4* is an essential mediator of cigarette smoke-induced pulmonary injury and emphysema and promotes NFκB activation, alveolar inflammation, oxidative stress, and apoptosis of alveolar septal cells [122]. As suggested in the German WES study, several genes involved in calcium ion and vitamin D-related biological activities are probably involved in sarcoidosis predisposition [113]. In sarcoidosis, macrophages activate the conversion of inactive precursor 25(OH)D to active vitamin D metabolite (1-25-(OH)2D3) and hypercalcemia is frequently observed in patients [123,124]. The hormonal form of vitamin D has immuno modulatory and anti-proliferative effects and is able to regulate the mTOR signaling pathway by stimulating expression of DDIT4, which facilitates the assembly and activation of the tuberous sclerosis complex (TSC)1/2 complex for eventual suppression of downstream mTOR activity through Rheb (Ras homolog enriched in the brain) [125]. Similarly, annexins (A11 or other forms) and a series of genes highlighted in the German WES screening encoding calcium-dependent enzymes (e.g., ACTN3), receptors (CNTNAP4), cadherins (FAT), and Ca2+ ion channels (e.g., CACFD1) suggest the importance of calcium metabolism in the regulation of immune processes and subsequent influence on mTOR activity and, probably, directly on autophagy. For instance, ACTN3 regulates serine/threonine-protein phosphatase 2B catalytic activity of calcineurin, a critical mediator of lysosomal calcium signaling and autophagy induction [126].

### 3.3. A Functional Link between Genomic and Transcriptome Analysis

The analysis of the transcriptome makes it possible to evaluate the over expression or, on the contrary, the loss of expression at the level of the messenger RNA of certain genes by comparing the samples of patients with those of control subjects. This method is complementary to genomic analysis techniques, GWAS and WES, but will not necessarily target the same genes, because a mutation in the structure of a gene is not necessarily correlated with an abnormality in its expression. The transcriptome will thus provide information on the pathways involved in the genesis of sarcoidosis, and will have to be performed on different types of biological samples, as well as in different clinical situations. An excellent review of the literature was published in 2017 on transcriptome data published since 2006 using either extracts of pathological tissue from sarcoidosis or preparations from BAL and, in some studies, the peripheral blood of patients [127]. All of these works suggest a deregulation of the gamma interferon (IFN-γ) driven STAT1 signaling pathway and a string Th1 immune response with the production of many STAT1 regulated cytokines. The involvement of JAK-STAT-dependent pathways was assessed by RNA sequencing analysis of skin-lesion samples from a patient with cutaneous sarcoidosis before and during therapy by Tofacitinib, a JAK-inhibitor [128]. Gene-set enrichment analysis showed abnormal activation of interferon-γ, tumor necrosis factor (TNFα), interleukin-6, STAT3, and mammalian target of rapamycin complex 1 (mTORC1) signaling before treatment compared with controls. Tofacitinib induced down-regulation of mRNA in the JAK-STAT-dependent pathways (interferon-γ and interleukin-6) as well in TNFα- and mTORC1-related pathways. Interesting approaches were to compare the transcriptome of self-limiting versus progressive lung sarcoidosis with fibrotic evolution, to search for sarcoidosis-specific gene signatures compared with tuberculosis, an infectious disease also characterized by granuloma, and/or to identify transcriptome signatures related to disease outcome [129]. A recent study focused on the transcriptional signature in monocytes from sarcoidosis patients and healthy controls by an RNA-sequencing screening of the poly-adenylated (or purified mRNA) fraction of RNA [130]. Among 2446 genes significantly differentially expressed in monocytes between sarcoidosis and controls, the authors identified a downregulation of genes involved in proteasome degradation and ribosomal pathways, and conversely, an upregulation of genes, such as, for example, ATP6AP1 (ATPase H + transporting accessory protein 1), CYBB (cytochrome b-245 beta chain), LAMP2 (lysosomal-associated membrane protein), PLK1 (serine/threonine-protein kinase or polo-kinase 1), and SERPINA1 (serpin family A member 1, encoding alpha-1 antitrypsin), implicated in phagocytosis and lysosomal pathways. Interestingly, these genes regulate autophagy through various processes, including proteasome activities, glycosylation and lysosomal acidification, or direct interaction with the mTOR hub, as shown for LAMP2 and PLK1 [131,132]. CYBB encodes an essential component of NADPH-oxidase, a membrane-bound enzyme complex that generates high amounts of microbicide superoxide and hydrogen peroxide that needs the small G-protein Rac1 in macrophages for full activation [133]. Transcriptome analysis, although complex, may in the future bridge the gap between constitutional genomic data and the results of functional studies. For instance, RNA-Seq analysis of CD14+ monocytes from non-hypoxic sarcoidosis patients showed enrichment for metabolic and hypoxia inducible factor (HIF) pathways, with abundance of HIF-1α/β in the center of granuloma [134]. The authors showed that this over production was linked to elevated IL-1β and IL-17, suggesting that increased activity of HIF-α isoforms regulates Th1/Th17 mediated inflammation in sarcoidosis, putatively related to local hypoxia within granulomas owing to the lack of vascularization. Hypoxia has been shown to exert a strong inhibitory effect on mTORC1 activity, with consequent inhibition of mTOR signaling [135]. Interestingly, this effect is mediated by DDIT4, a gene we found mutated in some sarcoidosis families, also known as REDD1 (regulated in development and DNA damage response 1) [105,136]. These data were assessed by a recent work showing that miR-7, a micro interfering RNA repressing DDIT4/REDD1 expression, was down-regulated upon hypoxia to consequently increase DDIT4/REDD1 and inhibit mTOR signaling [137]. This specific observation highlights the need for further RNA seq analysis, focusing on regulating micro RNAs that may regulate various genes involved in sarcoidosis-related pathways, and thus providing some links with genetic triggers of the disease [138]. All these data show that gene expression analysis must be considered as functional tests performed in various conditions, highlighting major processes involved in sarcoidosis, such as T-cell receptor, activation and signaling, macrophage polarization, Th1 and IFN immune responses, and the molecular machinery of autophagy serving phagocytes in many more membrane trafficking pathways, thereby regulating immunity to infectious disease agents.

## 4. Perspectives on the Clinical Utility of Genetics in the Management of Sarcoidosis Patients

The compilation of the data obtained by GWAS and, more recently, by whole exome sequencing shows how high the locus heterogeneity is. This does not facilitate a priori the screening of a particular gene for predicting the disease, for example, in familial forms. Undoubtedly, many more studies will be needed using either NGS gene panels, or more likely large-scale sequencing of the exome, possibly the genome. Nevertheless, a careful reading of the scientific literature on the function of the genes identified in genetic studies on sarcoidosis leads us to think that genetics will have a major utility (1) to identify specific pathways that may lead to innovative therapies for corticoid-resistant sarcoidosis and (2) to allow the characterization of genetic markers associated with particularly severe forms of the disease, either owing to persistence and progression to pulmonary fibrosis, or by lesions in many organs, or by the occurrence of opportunistic (viral, parasitic, or fungal) infections suggesting an immune weakness or autophagic of genetic origin. As described previously, these assumptions make sense when we analyzed the functional target (Rac1, mTOR) of the treatments currently used in advanced forms of sarcoidosis. This relationship between therapeutic and pathway can also be considered realistic for less specific treatments such as corticosteroids and leflunomide, which impact S6 kinase, a critical effector of mTORC1 [139,140]. Recent data showed that inhibitors (Tofacitinib and Ruxolitinib) of the JAK/STAT (Janus kinase/signal transducer and activator of transcription) signaling pathway were efficient for resolution of cutaneous sarcoidosis in a JAK2 mutation-positive patient [141]. JAK/STAT is activated by Rac1 and STAT activation has been shown to be regulated by mTOR in dendritic cells and macrophages [142,143]. Taken together, the functional analysis of therapeutic targets helps us to have a more integrated view of molecular disturbances in sarcoidosis. We firmly believe that sarcoidosis remains a complex disease, but is putatively classified into three main forms: one would be linked to de novo mutations with early expression in children and linked to severe mutations of linked genes to innate immunity and internal processes of autophagy; the second represents the common or rapidly resolving forms, associated with a combinatory effect of gene variants at various levels of the immune system and, more specifically, the regulation of T cell activation; and the third would be represented by hereditary forms and linked to gene alterations in the regulation of autophagy, particularly around the functional hubs Rac1 and mTOR, and probably calcium metabolism. Polygenic risk profiling to identify individuals who could benefit from the knowledge of their probabilistic susceptibility to sarcoidosis seems of no real use, except if the evaluation of risk may be related to environmental factors, such as occupational exposition to mineral particles. A recent gene–environment interaction study was performed in a Swedish series of sarcoidosis patients and showed that disease risk is modulated by smoking owing to genetic susceptibility linked to the presence of variants in genes and mainly encoding the B-cell expressed Fc receptor-like (FCRL) molecules and the interleukin 23 receptor [72]. In the previous ACCESS (A Case Control Etiologic Study of Sarcoidosis) dataset, significant correlations have been identified between some specific HLA class II alleles, such as DRB1*1101 and DRB1*1501 with insecticide and molds exposure, respectively [144]. Looking forward to this exciting, but complex research, future short-term challenges are to provide the clinician in charge of patients with biomarkers of severity, for early therapeutic adaptation.

## Figures and Tables

**Figure 1 jcm-09-02633-f001:**
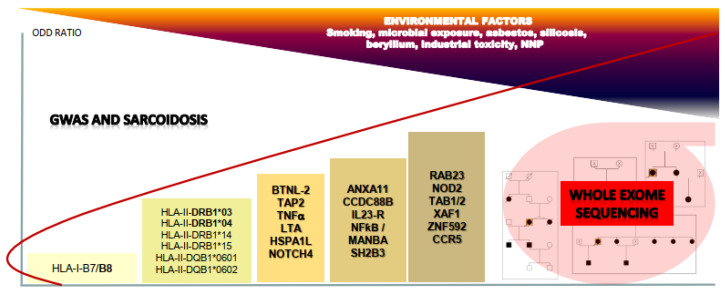
A representation of genes identified by genome wide association (GWAS) studies in sarcoidosis and the cumulative risk induced by the accumulation of variants closely associated with the negative impact of environmental factors. All the gene names are detailed in the text. The red line represents only a theoretical representation of the progression of the odds ratio to sarcoidosis when a patient accumulates polymorphisms in the different groups of genes. The knowledge provided by GWAS is complementary to the data from whole exome sequencing in families predisposed to sarcoidosis. NNP, nanoparticle.

**Figure 2 jcm-09-02633-f002:**
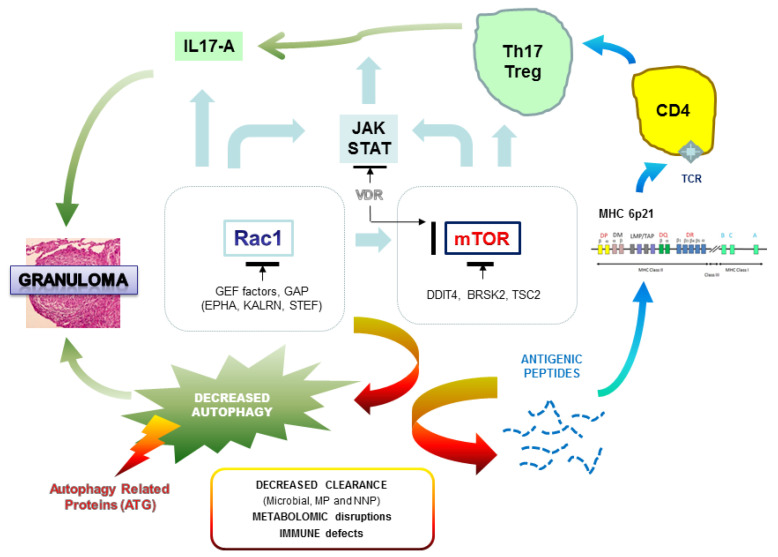
Functional hypothesis in the genesis of granuloma and relationship with deregulation of autophagy. MHC designates the major histocompatibility complex, VDR is the vitamin D receptor, and TCR is the T cell receptor. The black lines indicate a negative regulation. All the names of genes are detailed in the text.

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
