# Peer review of "Current Insights in Genetics of Sarcoidosis: Functional and Clinical Impacts"

_jcm, 2020, doi:10.3390/jcm9082633_

Round 1

Reviewer 1 Report

This is a submission of a significantly revised paper that was previously rejected.  The authors addressed many of the concerns of previous reviews regarding missing studies and inadequate treatment of large areas of research in genetics of sarcoidosis.  However, as stated in the original reviews, it is not clear what this paper contributes to the existing body of literature.  There are several similar review papers and even special issues of journals dedicated to these reviews.

Author Response

REVIEWER-1:

We agree with the reviewer that this manuscript is only a REVIEW but from my knowledge, no existing published works integrate the genetic background predisposing to sarcoidosis with updated data from whole exome analysis, functional pathways involved, a link with GWAS data and organ-specific expression of the disease and lastly data from transcriptome analysis. Our manuscript integrates all the WES studies published at this time and we have tried to focus some pathways highlighted by the different genes characterized, both by GWAS and whole exome sequencing. We hope that this work will be helpful in the frame of a large issue on the management of sarcoidosis and provide to the reader's the updated information on genetic predisposition to sarcoidosis

Reviewer 2 Report

In this review Alain Calender and colleagues summarize recent data on the genetic basis of sarcoidosis and integrate these data in a clinical model of sarcoidosis. This review is interesting gives an overview on the complex role of genetic factors in the immunopathogenesis of sarcoidosis. The authors did a good job to improve the current version of the review. I only have a view remarks:

  • Figure 1: if the antigen is beryllium it is chronic beryllium disease, not sarcoidosis! I agree that there are similarities, but in CBD there is a known antigen and a clear dependencie HLA restriction in T cell activation. Thus, CBD and sarcoidosis should be separated.
  • Figure 2: As before: it would help if the lines would be depicted as arrows.
  • I am not a native English speaker, but in my opinion, the manuscript requires language polishing:
    • line 46 “…incidence observed in the South European haplotype…”(?)
    • line 47 “observed as less as”
    • line 48 “…living in Europe…”
    • line 245 “…that play a critical functional…”
    • line 413: “…reported in sarcoidosis, he it allowed to stratify…” (?)
    • line 461: “…and negatively regulates negatively autophagy…”

Author Response

REVIEWER-2:

We thak the reviewer for the intersting comments on CBD

  • We have discussed more in details the differences between sarcoidosis and chronic beryllium disease by including two very recent works and specifying the close relationship of beryllium induced inflammatory process to specific HLA II alleles.
  • We made all the minor corrections needed in the text.
  • All the lines has been completed by arrows in in Figure 2

Round 2

Reviewer 1 Report

The paper is comprehensive in it's revised state and is much more clearly written but do not add to existing reviews already published.

This manuscript is a resubmission of an earlier submission. The following is a list of the peer review reports and author responses from that submission.

Round 1

Reviewer 1 Report

Alain Calender and colleagues attempt to summarize recent data on the genetic basis of sarcoidosis and to integrate these data in a clinical model of sarcoidosis. This review is interesting gives an overview on the complex role of genetic factors in the immunopathogenesis of sarcoidosis. I like this review; however, there are a few issues which need to be addressed.

General

  • In the abstract, the authors state that sarcoidosis “…belongs to the vast group of autoimmune disorders”, which is not commonly accepted. This reviewer also beliefs that sarcoidosis has an important autoimmune component, but is it really an autoimmune disorder in in a narrower sense?
  • Why should the pathogenic effect of ANXA11 on apoptosis limited to granuloma? The citations do not help as one is more than 30 years old and the other deals with ANXA1.
  • The authors should check their citations. In line 282 they refer to a German WES but the citation (95) describes the "ConsensusPathDB" but is not related with sarcoidosis. Do you mean Kishore et al. (DOI: 10.1007/s00439-018-1915-y, your citation 97)?
  • In line 403 the authors attempt to integrate the results from the genetic studies and immediately draw a line to a clinical application by defining biological or genetic markers. This is in my opinion a very clinical view. Shouldn’t we use the genetic data to identify pathways important in the immunopathogenesis of sarcoidosis? And then we might think on clinically applicable markers. I agree, given the divers clinical courses of sarcoidosis markers applicable to guide the currently increasing possible (but expensive) therapies are highly welcome. But can this delivered from WES or GWAS? In this case we do need much more clinical input into the genetic data. The authors do this later (which I love) as they discuss the link with current therapies with the pathways. However, they put it in a chapter starting with “Perspectives…”. Thus, in my opinion the authors should think to reorganize the parts 3.2 and 4 to get more logical and stringent.
  • Figure 2 is not that clear to me. The direction of the arrows (if they are arrows) is hard to see. The description is scarce (e.g. “peptides” I assume that you mean “antigenic peptides”). A at least short description of the scope of the figure is required.

Minor

  • From the author line (line 6) it seems that Thomas Weichart and Dominique Valeyre changed their positions ?.
  • The text requires language polishing due to some typos, grammar errors etc. E.g. line 40: “South Europa” should read “South Europe”; line 68/69: “…haplotype in a specific regions, so-called a candidate locus…” should read “…haplotype in a specific regions, a so-called candidate locus…”; the sentence in lines 74/75 seems to be incomplete; line 147: “…HLA alleles was associated…” should read “…HLA alleles were associated…”, line 170 “3 10-5” there is seemingly missing a sign between the number (x, ∙* or something like this), line 219: change “…critical functions role…”to “…critical functional role…”and others.
  • There are several changes in character styles.

Reviewer 2 Report

This is a well written manuscript that reviews genetic associations with sarcoidosis and also reviews some of the literature related to those mutations to try to understand their potential role in sarcoidosis. 

The title of the manuscript claims that the genetics of sarcoidosis could be a model for understanding gene-environment interactions. However these is little specific gene-environment discussions in the manuscript besides their mention of beryllium disease, smoking and the tentative association with Lofgren's. A discussion of the difficulties of doing gene-environment interactions would have been helpful. A reference that discusses this problem is Sarcoidosis Vasc Diffuse Lung Dis. 2008 Dec;25(2):125-32.HLA and environmental interactions in sarcoidosis.Rossman MD1, Thompson B, Frederick M, Iannuzzi MC, Rybicki BA, Pander JP, Newman LS, Rose C, Magira E, Monos D; ACCESS Group.

In the abstract's first sentence, the authors refer to sarcoidosis as an auto-immune  disease.  This is incorrect as it is not known whether sarcoidosis is an autoimmune disease or not. It would be better to refer to it as a hypersensitivity disease. Whether this hypersensitivity is to a self antigen (i.e. autoimmune ) or a external antigen is unknown. 

The discussion of specific HLA associations are mostly from case-control studies and the authors should make this clear.  One important manuscript that relates and a specific HLA to all sarcoidosis and not just a phenotype was not mentioned and should be cited. (HLA-DRB1*1101: A Significant Risk Factor for Sarcoidosis in Blacks and Whites, Milton D.RossmanBruceThompsonMargaretFrederick3MaryMaliarik4Michael C.Iannuzzi4Benjamin A.RybickiJanardan P.Pandey6Lee S.Newman7EleniMagira2BojanaBeznik-CizmanDimitriMonos2ACCESS Group*, AJHG Volume 73, Issue 4, October 2003, Pages 720-735.)

Whether the many genes identified by the GWAS studies that are reviewed in this manuscript will lead to novel therapies for sarcoidosis remains unknown, but the recent report that they cite of a JAK/STAT inhibitor is most encouraging. 

Reviewer 3 Report

Comments about the general content:

The review is focused almost entirely on findings in person's of European decent, and doe not mention the very likely etiological differences and unique genetic effects in other populations. Specifically, in the Abstract, line 14, the word autoimmune is used.  Given publications by Lareau et al (2017) and the CLEAR trials (Darke, 2013) among others, disease in non-european populations in not convincingly autoimmune.  This should be changed to reflect the current state of perspectives in the field.  

Also to the above point and the lack of comprehensiveness of this review, the references chosen are quite incomplete.  For example, on line 29, why were only the first 2 references cited here when there are a number of much more recent studies that support this statement, including 2 gene by environment association scans (Li et al 2014; Chen et al 2019) which are never even mentioned in the paper.

Similarly, line 50 cites one reference for heritability but doesn't even mention 2 of the largest epidemiological and family-based studies done in sarcoidosis: ACCESS and SAGA.  The authors discuss lack of "significant odd(s)-ratio risk" but don't clarify what "significant" means.  If they mean statistically, then yes, there are several; if they mean biologically, I would point to XAF1 (their ref 78) as well as HLA DRB1 0301 (their ref 27).

In general, the paper feels a bit dated.  Not only because more current references as well as all transcriptomic work is ignored, but even things like, on line 75, the authors state that genome wide scans are "heavy and costly" (not sure what heavy means in this context), but certainly GWAS is extremely cheap.  The cost of even the most comprehensive MEGA array is less than 40.00 a sample.

There is mention of 2 very small exam scans when, as mentioned above, but no reference to studies with 100s of exome, targeted and even WGS studies are mentioned (Levin 2014, Garman 2020 and others)

Again, Figure 2 seems very short-sighted in light of the obvious population specific, organ specific and disease course specific effects. 

Other specific points:

Line 16:  sarcoidosis is not a multidisciplinary field.  It is a multifactorial disease that requires the application of multiple disciplines.

Line 141, a recently published GWAS in ocular sarcoidosis should be referenced (Garman et al 2020)

References need to be check or alignment with the text, particularly beginning around #48.

On lines 215-218 a different font is being used.